# Improving Vitamin D Intake in Young Children—Can an Infographic Help Parents and Carers Understand the Recommendations?

**DOI:** 10.3390/nu13093140

**Published:** 2021-09-09

**Authors:** Ailsa Brogan-Hewitt, Tanefa A Apekey, Meaghan Sarah Christian, Rhiannon Eleanor Day

**Affiliations:** 1School of Clinical and Applied Sciences, Leeds Beckett University, CL615A Calverley Building, Leeds LS1 3HE, UK; ailsabroganhewitt@gmail.com (A.B.-H.); T.A.Apekey@leedsbeckett.ac.uk (T.A.A.); 2Department of Nutrition, Dietetics & Food, Monash University, Be Active Seep Eat (BASE) Facility, Level 1, 264 Ferntree Gully Road, Melbourne, VIC 3168, Australia; meaghan.christian@monash.edu

**Keywords:** vitamin D intake, infants and children, infographic, qualitative research

## Abstract

Vitamin D deficiency is a serious public health issue in the United Kingdom. Those at increased risk, such as pregnant women, children under 5 years and people from ethnic groups with dark skin, are not all achieving their recommended vitamin D. Effective vitamin D education is warranted. A qualitative study was undertaken to evaluate the acceptability and understanding of a vitamin D infographic, developed using recommendations from previous research. Fifteen parents/carers, recruited through local playgroups and adverts on popular parent websites, participated in focus groups and telephone interviews. The majority were female, White British and educated to degree level. A thematic analysis methodology was applied. The findings indicated that understanding and acceptability of the infographic were satisfactory, but improvements were recommended to aid interpretation and create more accessible information. These included additional content (what vitamin D is; other sources; its health benefits; methods/doses for administration and scientific symbols used) and improved presentation (eye-catching, less text, simpler language, more images and a logo). Once finalized, the infographic could be a useful tool to educate families around vitamin D supplementation guidelines, support the UK Healthy Start vitamins scheme and help improve vitamin D status for pregnant and lactating women and young children.

## 1. Introduction

Vitamin D is a hormone precursor that is synthesized in the skin during UVB radiation from sunlight exposure [1]. The primary circulating vitamin D metabolite in the body is 25-hydroxyvitamin D [25(OH)D] and is used as an indicator for vitamin D status [2]. Most vitamin D is obtained through sunlight exposure [1,3]; dietary sources only account for between 10 and 20% of the body’s total store [4], as few foods contain vitamin D naturally and fortification of foods in the UK is limited [2,5]. Vitamin D is a key regulator of calcium and phosphorous metabolism in children and adults [6] and is critical for bone health [7], through the mediation of osteogenic-related mineral absorption and utilization for skeletal development and maintenance [1]. Furthermore, due to the presence of vitamin D receptors expressed in almost every tissue and cell, there have been a substantial number of investigations into the effects of vitamin D that are extra-skeletal [8,9,10,11,12,13,14,15]. Moreover, there are current investigations into the impact of vitamin D on COVID-19 [16], with recent systemic reviews and meta-analyses revealing that low serum 25(OH)D is significantly associated with a higher risk of COVID-19 infection and severity [17,18].

Vitamin D deficiency (VDD) occurs when 25(OH)D concentrations fall too low, for example below 25 nmol/L can lead to poor bone health, which can present as bone pain or skeletal deformities, such as osteomalacia and rickets [19]. In children, severe VDD can cause hypocalcaemia (low levels of calcium in the blood), associated with seizures, tetany, heart failure [20,21], skeletal degradation and rickets [3,22], with osteomalcic leg bowing, as well as muscle weakness and delayed infant development [3]. Rickets and hypocalcaemic complications remain a serious health issue and can cause infant mortality, especially in the Black, Asian and Minority ethnic (BAME) groups in the UK [3]. According to the Scientific Advisory Committee on Nutrition (SACN) [23], VDD is more prevalent during periods of rapid growth, and therefore children under five are at increased risk. Furthermore, prolonged vitamin D insufficiency has been associated with further health problems in adulthood, including cardiovascular disease, diabetes and autoimmune diseases [24,25], with implication for future costs to the UK’s National Health Service (NHS). There has been much discussion and debate recently about the optimal 25(OH)D serum concentrations for an increasing range of health issues [7,26,27,28]. It is clear, however, that severe VDD is a long-held and ongoing concern for the UK population, evidenced by a recent resurgence in rickets [29,30,31,32], increasing hospitalization rates of children related to lack of vitamin D [4,31] and increasing costs of vitamin D prescriptions and tests for children with VDD in primary care [33]. Furthermore, VDD in UK children has increased in recent years [33], with 19% of children aged 4–10 years and 37% aged 11–18 years, having 25(OH)D below the deficiency threshold, with BAME groups being by far the most affected [3]. Further still, one in four young children are not obtaining the recommended vitamin D intake essential for their musculoskeletal development [34], with children under four years old only receiving less than one-third of their dietary vitamin D recommendation [35,36]. There is also increased risk during pregnancy when sufficient vitamin D is crucial in aiding the absorption of calcium, when demands for the growing fetus are higher [37]. It has been reported that 45% of pregnancies in England are unplanned and many women of childbearing age do not seek preconception care [38], which poses an additional risk to this group, as they may not be aware of the importance of taking vitamin D supplements. In addition, the risk of osteomalacia is more prevalent in pregnant women, due to lack of adequate vitamin D [23,39]. Furthermore, children who are exclusively breastfed are also at risk, as the infant intake relies on the mother’s vitamin D status [40]. In response, the UK has adopted a supplementation policy [3], which recommends that infants under one year should have a safe intake in the form of a daily supplement of 8.5 to 10 micrograms of vitamin D. This includes those infants that are exclusively and partially breastfed, but excludes infants consuming more than 500 mL of formula per day, as infant formula is fortified. Children aged one to four years and everyone above this age, including pregnant women, should have a reference nutrient intake (RNI) of 10 micrograms of vitamin D daily, especially throughout the winter months of October to April. Vitamin D can be provided free of charge to those on a low income through the Healthy Start Scheme: a UK government initiative that provides free vitamin supplements for all low-income families, women throughout pregnancy and for children from 6 months to 5 years [41]. The UK vitamin D supplementation policy has, however, been shown to be poorly implemented [42] and not fully effective [3], as evidenced by the mean daily vitamin D intake for children still below the RNI of 10 micrograms per day [35,36]. Uptake of vitamin D supplementation has been reported to be low in pregnancy [43], in women who breastfeed, as well as in children under five years [44,45]. Furthermore, mothers have not been shown to be motivated to take supplements themselves, nor give them to their children [41], suggesting that they are not aware of the benefits. Uptake of the Healthy Start scheme has also been reported to be poor across the UK; data by the Healthy Start Unit estimated that the Healthy Start scheme would see a registration of 80%; however, the estimated uptake of the vitamin D supplement component was then extremely low (2 to 4%) [46]. Reasons for this may include poor awareness and access to the scheme, compounded by poor distribution of supplements and promotion of the scheme by Health Care Professionals (HCPs) [41,47]. Furthermore, previous studies have indicated lack of parental knowledge and awareness of the current vitamin D recommendations [13,43,48,49,50], with many parents not being given adequate vitamin D information [47,51]. In addition, the recommendations for infants have been found to be difficult to interpret [42,47,52], and parents wanted simpler, easier to interpret guidelines, at different stages of their child’s development [47]. Moreover, research has shown that parents and carers of young children who were previously unaware of the recommendations, once made aware, would consider giving vitamin D to their child [48].

There is therefore a recognized need for effective and impactful public health education on vitamin D recommendations especially in at risk groups. One of the most innovative and engaging methods for presenting health information is through a design technique called an ‘infographic’. Infographics are visual presentations of information, that use elements of design to display content; by expressing complex messages to viewers in a more interpretable way, to enhance their comprehension to those with varying health literacy levels [52,53]. Furthermore, they are thought to help make information more appealing, accessible, attention-grabbing and memorable [52]. In addition, they can build a bridge between health professionals and lay people, by helping important points be better understood and acted upon [54]. An infographic succeeds if attention is drawn to it, people share it within their networks, we can understand the information presented and this translates to people actually considering behavior change [54].

To the researchers’ knowledge, this type of research is the first of its kind, with regard to testing an infographic displaying the current UK infant vitamin D recommendations, and thus provides unique information to inform the development of a valuable tool to effectively educate the community around vitamin D in the early years. This study therefore aims to develop and test the acceptability and understanding of an infographic displaying vitamin D recommendations, with a group of parents or carers of children aged under 5 years, as an intervention to improve vitamin D status in young children.

## 2. Materials and Methods

### 2.1. Development of the Infographic

The infographic was developed in response to findings from previous research [13,42,43,47,48,50,51,55] that identified that there is poor awareness of vitamin D recommendations, which appears to be a barrier to vitamin D uptake, there is a need to aid education and improve awareness around vitamin D recommendations and there is a lack of informative and accessible vitamin D educational materials for parents. This presented an opportunity to develop an infographic displaying key information surrounding current UK infant vitamin D recommendations. The infographic was developed and tested using the six essential Steps for Quality Intervention Development (6SQuID) [56]: (step 1) defining and understanding the problem and its causes; (step 2) identifying which causal or contextual factors are modifiable: which have the greatest scope for change and who would benefit most; (step 3) deciding on the mechanisms of change; (step 4) clarifying how these will be delivered; (step 5) testing and adapting the intervention; and (step 6) collecting sufficient evidence of effectiveness to proceed to a rigorous evaluation.

A panel of experts in the area of child health and lay advisors was established, comprising the research team; a public health specialist; a midwife; an academic in child health and two parents of children aged under 5 years. Together, they reviewed the recommendations from previous research and co-developed the infographic, achieving steps 1 to 4 of 6SQuID. The methodology for testing the infographic was also agreed and refined to achieve step 5. A preliminary infographic was then developed for pregnant women and parents of children aged under 5 years, by the media team at the university. The vitamin D infographic is included as a supplementary file (Appendix A). It demonstrates the vitamin D recommendations broken down by different stages of the child’s development/feeding, as recommended by the participants of our previous research, who felt that this would make them easier to interpret [47]: pregnancy; breastfeeding exclusively (0–6 months); formula feeding (0–6 months); weaning (6 months–1 year) and for the child aged 1 to 4 years.

### 2.2. Testing the Infographic

A qualitative approach, using focus groups and interview methodology, was adopted. It was thought that focus groups would facilitate rich discussion and in-depth feedback [57,58] on the appropriateness, interpretation, usefulness and applicability of the newly developed tool. It would give an opportunity for participants to discuss their views in their own words and uncover important issues that might not have been previously considered. A focus group topic guide was devised to explore the following: understanding of the vitamin D recommendations at different stages of their child’s development, acceptability and perceptions of content, appearance and delivery and further recommendations for improvements. The focus group topic guide was piloted with a small group of parents initially; following this, minor changes were made to wording for clarification. The focus group topic guide is included as a supplementary file (Appendix A).

#### 2.2.1. Participants and Recruitment

The eligibility criteria for participation in this study included pregnant women and/or parents and carers over the age of 18 years, who care for a child up to the age of four years old, living in a city in the North of England. Those under the age of 18 years, or those who were not a parent/carer or pregnant and those who could not understand written English well enough to interpret the infographic, as well as those who did not provide consent, were excluded from this study. The target age range was selected because children under five years are at increased risk of vitamin D deficiency [23] and vitamin D recommendations have been reported to be more complex for children under this age. This study was advertised (via a recruitment poster) on the parent-centered website Mumbler and within community centers and playgroups serving low-income and diverse ethnic areas across the city. Parents or pregnant women who were interested in participating in the focus group were requested to contact the researchers directly. The sample comprised fifteen parents/carers (grandparents) of children under the age of 5 years.

#### 2.2.2. Procedure

The participants were given an information sheet, a consent form and a demographic questionnaire to complete prior to participating in this study. Any questions about this study were addressed prior to participation. The interviews and focus groups were digitally recorded after written consent had been obtained from the participants. Individuals were also assured that they would not be identifiable in the report of findings. The infographic was displayed during each focus group for reference and a digital copy was emailed beforehand to participants involved in the telephone interviews. The focus groups and telephone interviews were conducted by one female researcher (ABH, Master of Clinical Nutrition), having been trained in qualitative research methods, and lasted approximately 30–40 min. No one else was present during the interview/focus group except the researcher and participant(s). They were recorded using a digital audio-recorder. Field notes were also taken at the time of focus group. Fifteen participants took part in four focus groups and two telephone interviews, carried out in November 2019. These consisted of one group of five participants, one group of four participants, two groups of two participants and two telephone interviews. The focus groups were carried out in playgroups and places where the participants worked within the city, which were deemed convenient locations with regard to accessibility, comfort and where permission had been obtained. All digital data such as consent forms, demographic information, audio recordings and transcripts were stored in a password-protected file, which only the research team had access to. Physical participant data were stored in a lockable filing cabinet. Ethical approval was provided by the Leeds Beckett University ethics review committee (reference number 57801).

### 2.3. Data Analysis

The recordings of interviews and focus groups were listened to for familiarization and transcribed verbatim by ABH, by listening to the audio-recordings and typing into Microsoft Word documents. This was carried out within 7 days post interview. Additional information from field notes was added to the transcripts. The transcripts of the interviews and focus groups were anonymized; participants’ names were replaced with unique identifiers and identifying details (e.g., names of individuals) or any sensitive data referred to within transcripts replaced with codes. A thematic analysis methodology was applied, guided by Braun and Clarke [59]. Lincoln and Guba’s [60] six-stage criteria for trustworthiness approach was also used within the data analysis, to ensure consistency and cohesion [61]. Reflexivity journaling was also carried out after each session by ABH to help provide context and understanding of this study and to take learning from and improve subsequent sessions [62]. In addition, at each stage of analysis, a self-critical account of the research process was documented [63]. The analysis was conducted over a number of stages: members of the research team read and familiarized themselves with the content of the transcripts. Based on this, a coding framework was developed (by ABH), which was derived inductively from thematic areas of interest within the data. The coding framework was refined and agreed amongst the research team (ABH, RED, MSC, TAA) and applied to the original transcripts to extract major themes. Firstly, descriptive words were found for each participant and coded (by ABH), and then the second level of coding was carried out, focusing more on the interpretation of the codes [64]. Coded data were categorized and collated into themes (manually) (by ABH); once the inductive coding process had been carried out, the codes were grouped into sub-themes and then the main themes were identified [65]. The emergent themes generated were agreed within the team (ABH, TAA, RED, MSC). Data saturation of these codes was achieved after the first four focus groups, as there were no new codes and therefore themes identified after this point [66]. Appropriate quotations were also selected to represent each theme. Microsoft Excel was used to calculate basic descriptive statistics from the demographic questionnaire, such as counts and percentages and to create tables. The percentages are presented as whole numbers. The Consolidated Criteria for Reporting Qualitative Studies (COREQ-32 item checklist) [67] was also applied to the report of findings.

## 3. Results

### 3.1. Description of Participants

Table 1 demonstrates the demographic characteristics of the participants. The majority were female, of White British ethnicity, aged between 25 and 34 years, mothers or carers of at least one child under four years. Most of the sample (80%) had at least an NVQ Level 4 qualification (Level 4 is indicative of a greater understanding and a higher level of learning than that gained through core modules at secondary education level in England, Wales and Northern Ireland), and just over half had a degree. This is significantly higher than the city average (46.9%) [68] and the national average (44%) for an NVQ Level 4 qualification or above [69].

### 3.2. Findings from the Focus Groups and Interviews

The findings have been summarized into the following main themes: general awareness of vitamin D; understanding of the vitamin D infographic; acceptability of the vitamin D infographic; barriers to using the vitamin D infographic and recommendations for the vitamin D infographic. The content of each is explored in more detail below. Relevant quotations from participants are presented to illustrate key points, with interview/focus group number and participant number labelled in brackets. There were no differences in perceptions of participants by age or education level.

#### 3.2.1. General Awareness of Vitamin D

The following sub-themes were developed from the analysis of the discussions: awareness of sources of vitamin D not displayed (on the infographic) and awareness of infant vitamin D recommendations. Many participants had some awareness of vitamin D in the context of obtaining it through sunlight exposure, with a few acknowledging the difficulties with getting enough, particularly in the winter. Whilst some participants stated that they had seen or used foods fortified with vitamin D, general awareness of food sources containing vitamin D and their contribution to overall vitamin D needs was poor. These points are illustrated with the following quotations:

*“You can get sunlight as well but it’s not a thing here [in the UK]”* (Interview participant 1)

*“I see it printed on a lot of yoghurts, is it like green veg?”* (Focus group 1, participant 3)

*“If it’s in their diet and in their milk, it’s all a bit confusing isn’t it?”* (Focus group 4, participant 3)

*“…I still don’t make an effort to take supplements I just hope that somehow I might be getting what I need from my diet”* (Interview participant 2)

Whilst a few participants mentioned that they had received vitamin D information during pregnancy and after the birth, they had not seen the information broken down by developmental stages before and perceived that this was a novel approach. Furthermore, the information was completely new to some participants, for example the need for vitamin D for the mother during breastfeeding.

*“I didn’t know all the information on there, I didn’t know the mother was supposed to take some. I didn’t know there was a difference between breast milk and formula”* (Focus group 3, participant 1)

#### 3.2.2. Understanding of the Vitamin D Infographic

The following sub-themes emerged from the discussions: understanding of what the infographic was telling them; understanding of language used; understanding of methods, doses and symbols for vitamin D supplementation; fear of overdosing and need for reassurance by a health professional. Many stated that they understood what the infographic was telling them in relation to the vitamin D recommendations for supplementation at different development stages for the child and perceived that the language used was generally “clear”. However, for others, it was not immediately clear what the infographic was depicting and perceived that it took some deciphering, as illustrated through the following quotations:

*“Yes, it’s trying to advise the recommended dosage for different scenarios at different stages”* (Focus group 3, participant 1)

*“It is definitely quite difficult…to read it. I’m here to read that, so I’m actively reading it, but I don’t think at a glance at it in a doctor’s surgery or something, you wouldn’t know instantly”* (Focus group 3, participant 1)

Furthermore, some found the information on vitamin D methods of supplementation and doses difficult to interpret and the majority of the sample did not recognize the micrograms symbol ‘μg’. Some had previously seen it on vitamin packaging, but others had never seen it before, as illustrated with the following examples:

*“The signs, I don’t know what they mean?”*(Focus group 4, participant 3)

*“Especially with this symbol here, I’ve seen it on vitamins [sic], but you would always step back and wonder, whereas all they have to do is get a vitamin D supplement”* (Focus group 2, participant 1)

Further still, a few expressed concern over the potential to overdose children with vitamin D, demonstrating a lack of clarity on vitamin D intake, as illustrated here:

*“Can you overdose on vitamin D? Because if you’re not sure what’s in the milk, you don’t know how much they’re getting?”* (Focus group 4, participant 3)

Moreover, a few participants highlighted that they would still like more information/support about vitamin D from a health professional on reading the infographic, perhaps indicating a lack of confidence in or understanding of the information, as illustrated here:

*“If I look at the infographic, I would like GP support or someone that has any connection with health, that could help more [sic]”* (Interview participant 1)

#### 3.2.3. Acceptability of the Infographic

The following themes emerged from the discussions: acceptability of information displayed; missing information; acceptability of presentation and trustworthiness of information. Acceptability of the infographic was generally satisfactory; the majority of the sample liked how the recommendations were illustrated for different developmental stages, as illustrated with the following quotations:

*“…I like the stages. And obviously that it’s got reliable information about all the stages in one place”* (Focus group 2, participant 1)

Much of the sample thought, however, that the following information was missing from the infographic: ‘why’ vitamin D is needed; where it can be found and the health benefits. Furthermore, it was expressed that the inclusion of this would encourage them to follow the advice. This is highlighted with the following quotations:

*“It doesn’t actually tell you what the benefits of vitamin D are? It just tells you how much people should be taking”* (Focus group 3, participant 2)

*“It’s not actually giving you any practical information about what you do to get your vitamin D. Do you get it naturally? You know, what supplements? Are you saying a pill?”* (Focus group 4, participant 2)

*“It’s very informative but it doesn’t actually say why you need it. I mean we all just do what we’re told with this kind of stuff. If we have to have it in pregnancy, we just have it in pregnancy”* (Focus group 2, participant 2)

Much of the sample perceived that the infographic was “eye-catching”, “straightforward” and “practical”. A few thought that it looked “professional” and liked the “scientific” background images on the infographic and the images accompanying the text. However, for others, the color, images and choice of text font were not appealing and there was too much text presented, as illustrated here:

*“It looks old fashioned, I don’t know whether that’s the font at the top. I don’t mind the body text as I think that’s quite a modern font, but I think the headline, and the orange/yellow makes it look quite dated”* (Focus group 2, participant 1)

*“I don’t like the amount of text”* (Focus group 3, participant 2)

Furthermore, many thought that acceptability of the infographic would improve if it was clear that it was from a trustworthy and reliable source and that an association with a common health care provider, such as the NHS, would improve this, as illustrated here:

*“…it doesn’t say who it’s recommended by and doesn’t confirm the credentials”* (Focus group 3, participant 2)

#### 3.2.4. Barriers to Using the Vitamin D Infographic

Perceptions of barriers to using the infographic effectively were also discussed by some participants. It was thought that issues such as the language used might hinder understanding of the infographic, as well as being able to remember the information. Further, it was also perceived that the cost of vitamin D supplements might deter people from purchasing supplements. Awareness of the UK Healthy Start scheme was not apparent in this sample. Furthermore, it was highlighted that some might not know what vitamin D supplements actually look like and thus would not know what to look for, as illustrated with the following examples:

*“Parents in EAL’s (English as an additional language) or say if it was in the middle of Bradford or Birmingham, a lot of them might not be able to read it as easily”* (Focus group 3, participant 2)

*“The thing is, I wouldn’t know what I’m looking for, for a vitamin D supplement”* (Focus group 1, participant 1)

#### 3.2.5. Recommendations for the Infographic

The following themes emerged from the discussions: need for further information (additional content); improvements to presentation; preferred format and delivery; accessibility to and promotion of the infographic and endorsement of information. Recommendations for improvements to both the content and presentation of the infographic were made by some. These included displaying information about other sources of vitamin D; including sunlight and food/drink sources, why vitamin D is important and the health benefits of vitamin D. Moreover, some perceived that it would be useful to know what the recommendations for children over 4 years were. These points are illustrated with the following quotations:

*“Maybe just a paragraph at the beginning explaining why vitamin D is important. You know, what’s the point in reading it if you don’t know why?”* (Focus group 4, participant 4)

*“It may be useful if you say the importance of taking vitamin D and why taking vitamin D for those stages of life for the child”* (Interview participant 1)

Furthermore, some expressed that there was a need for more succinct, easier to interpret information, written in plain English. There were also recommendations for improved display of information, such as in columns or tables, a brighter and more appealing color, a more modern font and more images, as illustrated with the following quotations:

*“…so like that first one, a vitamin D supplement containing 10 micrograms for mother daily could be, ‘mum needs one 10 microgram supplement daily’”* (Focus group 3, participant 2)

*“…if you could simplify it in a set of columns where you could just tick, if I need to take one thing away what should that be? If my child is this age you need to do this”, “It’s just making it easy to take away what I need to do”* (Focus group 2, participants 1 and 2)

*“…it might help if there was a picture of a sun, to show them where it comes from?”* (Focus group 4, participant 1)

There were also recommendations for clearer information on what vitamin D supplements both look like and which doses are needed for their children, and that the infographic should remind parents to check the labels of supplements to make sure that the dose is adequate to meet the recommendations for their child, as illustrated with the following examples:

*“Maybe if it showed you the supplements, what they look like?”* (Focus group 1, participant 2)

*“You may think if you give your child a multivitamin they are ok. But how much is there in it?”* (Focus group 4, participant 3)

Many participants also made suggestions for their preferred delivery of the infographic. Many stated that they would prefer the infographic to be made available in a printed hard copy, but there were also suggestions for it to be available by email or digitally (that could be downloaded or screen-grabbed for a picture). Other examples included, providing a copy in the child’s progress record (red book) or as a small laminated card to fit a wallet, similar to the one provided for meningitis; making it look more “permanent” and memorable. Many stated that they would expect to see the infographic in health care locations such as General Practitioner (GP) surgeries, hospitals, pharmacies or during their child’s review or midwife appointments. Other suggested locations included schools, nurseries, children’s centers, sports centers, swimming pools, village halls, cafes, supermarkets, or “anywhere you find mums and dads.” Additionally, providing a copy in the packs of information women receive when they are pregnant and developing a public health campaign about vitamin D, were thought to be useful strategies. Many of the participants also suggested that advertising it on social media could be a good way of promoting the infographic. It was also mentioned that it should be available in different languages. These points are illustrated with the following quotations:

*“When you go to see the midwife for a booking, that might be really useful to highlight it”* (Interview participant 2)

*“I think social media would have more impact on people’s lives, because everyone nowadays has a social platform”* (Interview participant 1)

Furthermore, there was a feeling that they would be more likely to trust the information if it was available in a medical location and showed that it was endorsed by a health care organization, such as the NHS. Further still, some of the sample felt that there should be information on where to go if people need to ask more questions, such as contact information/website address, and that health care practitioner support should be available to help clarify things if needed, as illustrated here:

*“…if it had an NHS logo stamped at the bottom, I would be like, that is absolutely (trustworthy)…I’d feel like it’s been validated”* (Focus group 2, participant 1)

*“Is there like, anyone else you can ask the questions? Let’s say it was confusing say between swapping from formula, could you not put on the other people to ask about that? [sic]…If there was a website at the bottom of it with more information”* (Focus group 1, participant 3)

## 4. Discussion

The findings revealed that the sample had some awareness of vitamin D in the context of obtaining it from sunlight, but there was a lack of awareness of how vitamin D needs can be adequately met, with some perceptions that dietary sources alone are sufficient to meet vitamin D requirements. Whilst many understood what the infographic was reporting, some found it unclear and difficult to interpret, particularly regarding vitamin D doses and symbols. Acceptability of the infographic was generally satisfactory, with the majority liking the breakdown of the vitamin D recommendations by developmental/feeding stage, but there were mixed reviews on the presentation of the infographic. Perceived barriers to using the infographic effectively included the language used, remembering the information and the cost of vitamin D supplements, as well as not knowing what the supplements look like. Several recommendations for improvements to both the content and presentation of the infographic, as well as how it would be delivered to parents and pregnant mothers were made to improve its accessibility and acceptability.

Despite efforts to display the vitamin D recommendations in a succinct and simplified way (the need for which was highlighted in previous research), it was clear that some still struggled to interpret the information easily. This highlights the need for further improvements to make the infant vitamin D recommendations easier to interpret and accessible to those of varying degrees of health literacy. The addition of some information around what vitamin D is, the different sources (sunlight, dietary) and why it is needed, would help parents contextualize vitamin D and its importance, and facilitate interpretation of the infographic, as it was clear that there was not a great awareness around vitamin D generally. Furthermore, the participants of this study were above average education level and it could therefore be argued that awareness around vitamin D recommendations for their children could be even lower in those less educated, furthering health inequalities [71]. This also further supports the evidence that not all parents are receiving adequate information about the need for vitamin D supplementation in children [47,51]. This therefore presents a need for further clarification of how vitamin D needs can be met and the need for vitamin D supplementation for young children to be further emphasized on the infographic.

The findings also indicated that there was reported confusion surrounding the different methods of supplementation and doses required at different stages, including interpretation of symbols (μg) for vitamin D supplement dosage. This has been highlighted as an issue in previous studies [43,47,49,50]. Furthermore, lack of understanding of scientific numerical data has been found to be a barrier to acting upon health advice previously [72,73,74]. This information would therefore need to be made clearer on the final infographic, with further explanation of scientific units. Additionally, this clarification could help overcome some of the confusion highlighted in this study regarding vitamin labelling, dosing and fear of overdosing. The addition of all the recommended information by participants would need to be carefully considered, however, as adding more information may only create more confusion. This was both highlighted in this study and within previous research [75], where infographics depicting health information with too much text, visually overwhelming information and excessive data have been found to be less effective. The addition of more images to the infographic as recommended in the current study could be a valuable way of facilitating interpretation, as research shows that images are a faster and more effective way of communicating information than text alone [76]. Moreover, it has been suggested that the use of simple wording supporting imagery is equally important, as people with low literacy skills may not be able to apply context without them [77]. Therefore, in order to ensure the infographic displays easy to digest information, reducing the amount of text and increasing the use of illustrations could improve its acceptability. Furthermore, the addition of a logo to indicate that the infographic was endorsed by a reliable organization would aid trust in the information and thus increase acceptability [73,78].

The findings also indicated that some of the sample felt that the cost of vitamin D supplements could prevent others from adhering to the recommendations on the infographic, with no apparent awareness of the UK Healthy Start scheme within the sample. Research [41] has indicated that the scheme is generally not well known within the population. The addition of information about the Healthy Start scheme to the infographic, such as a website address, might help facilitate access to vitamin D for those on low incomes. Furthermore, it may also help promote the scheme more widely, as well as prompting health care professionals to discuss it with parents, as advised by the National Institute for Health and Care Excellence [79], since research has previously indicated a low awareness of the scheme amongst health care practitioners [46]. There was also a perception that language could be a barrier to interpreting the infographic and following its recommendations. This could be mitigated by translating the text and utilizing more illustrations [53]. However, language discrepancies can result in significant communication errors [80], and therefore the thoughts and opinions of those whose first language is not English should be considered when translating the infographic.

A minor theme that emerged in the findings was the need for further advice around vitamin D recommendations from a health professional on reading the infographic. This perhaps highlighted a lack of confidence and understanding in the information provided, but also further illustrated that these recommendations for young children are indeed complex, compared to other groups, and this could be contributing to the difficulty with their interpretation, and thus overall lack of knowledge on vitamin D supplementation [42,52,54,81]. The importance for more simplified, easy to follow and accessible information is clear. Furthermore, there is some evidence of low levels of awareness of the importance of vitamin D supplementation in early childhood amongst health care practitioners [46,82], and thus some health care practitioners may not be adequately informed to give vitamin D advice to parents. The infographic thus needs to be clear enough so that it can be delivered as standalone information and could be used to support health care practitioners, when delivering verbal information to pregnant women and parents. Once this is achieved, the infographic has the opportunity to serve a dual purpose of educating parents and health care professionals [73].

The findings indicate that sufficient evidence was collected to inform the development of a final vitamin D infographic. The need for further evaluation with a larger group of parents is warranted, as outlined in step 6 of 6SQuID [56]. Based on the findings of the current study, the following recommendations have been suggested for the final vitamin D infographic: (1) Recommendations for content: also include information on what vitamin D is, other sources of vitamin D (dietary and sunlight), why we need vitamin D and its health benefits (to highlight its importance); the different methods for vitamin D supplement administration; vitamin D recommendations for children over the age of 4 years; information about the Healthy Start scheme and a website/contact information for more information about vitamin D. (2) Recommendations for presentation: ensure that it is eye-catching and stands out with appealing colors; simplify and reduce text, use illustrations/images to support interpretation; define and explain scientific units used, for example provide the word ‘micrograms’ alongside its symbol; ensure that is has a logo to endorse its credibility. (3) Recommendations for delivery and availability: make the infographic available in a variety of formats, such as printed hard copy/email/digitally. It could be included in the child’s development record (red book) additionally; make the infographic available in a variety of locations that parents and pregnant women visit frequently, such as GP surgeries, schools, nurseries, pharmacies, children’s centers; make hard copies of the infographic available to parents and pregnant women at routine midwife and health visitor appointments. It would be beneficial to also make it visible/available within a variety of other locations that families visit, such as sports centers, swimming pools, cafes and supermarkets; use social media to promote the infographic and increase accessibility by making it available in other languages.

To the researcher’s knowledge, this type of research is the first of its kind, and provides unique information to inform the development of a valuable tool to effectively educate the community around vitamin D recommendations in the early years. Another strength of this study was that the qualitative approach (focus groups) facilitated honest and open, in-depth discussions about the infographic and participants felt comfortable to critique the infographic. In addition, due to the nature of this research method, opportunity for the participants to seek clarification on questions asked was able to be provided, which means that it was likely that the data collected generated genuine feedback specific to this group [58]. Therefore, from the feedback provided, the research team are able to make improvements to the infographic to ensure it is accepted and understood by the intended reader. In addition, a reflexive journal and triangulation of the data collected, such as field notes, transcriptions and the original recordings, contributed to reducing researcher bias, a limitation of qualitative research. Moreover, to minimize bias, the original data and themes were peer-reviewed by the research team, further adding to the robustness of this study and all the study findings (both major and minor themes) presented with quotations from a variety of participants used to illustrate findings.

The main limitations of this study were the small number of participants and the lack of generalizability of the sample. Whilst this study aimed to gain a representative sample in terms of ethnic and socio-economic diversity, the sample was, however, 93% White British, compared to the city average of 81% [83]. In addition, the majority of the sample were well educated and had at least an NVQ Level 4 qualification (80%), almost double the city [68] and national averages [69]. Therefore, the findings may be transferable to similar settings, but not to other ethnic groups, those from lower socio-economic groups and those of a lower educational level. Efforts to recruit individuals from minority ethnic backgrounds, with different first languages and those from more socially economically diverse areas, as well as a larger sample, need to be made for further evaluation of the infographic [79]. Establishing contacts within the Local Authority’s public health team, working with diverse communities, schools, nurseries and playgroups for example, to help disseminate study information might help facilitate this.

## 5. Conclusions

The findings from this study indicated that general understanding and acceptability of the infographic were satisfactory, but there were recommendations for improvements to content, presentation and delivery of the infographic to improve overall accessibility and acceptability. The recommendations from this study can be used to develop a final vitamin D infographic to be evaluated with a larger sample of parents and pregnant women. Once finalized, it could be a useful tool to support vitamin D education in the community, support the UK Healthy Start scheme and help improve vitamin D status in both pregnancy and in children under the age of 5 years.

## Figures and Tables

**Table 1 nutrients-13-03140-t001:** Characteristics of participants.

Characteristics	*N*	%
Gender		
Male	2	13
Female	13	87
Total	15	100
Age of participant (years)		
25–34	6	40
35–44	4	27
Over 45	5	33
Total	15	100
Age of child(ren) (years)		
Under 1	2	13
1	3	20
2	10	67
3	5	33
4	1	7
Total	21	140
Relationship to child		
Mother	8	53
Father	2	13
Grandparent	5	33
Total	15	100 *
Ethnic background		
White-British (English, Welsh, Scottish, Northern Irish)	14	93
Mixed-Other	1	7
Total	15	100
Highest Qualification **		
Level 2 (Level 2 NVQ, CSE—grade 1/GCSE grades A *–C, Level 2 award, Level 2 certificate, Level 2 diploma, Level 2 EOSL, Level 2 national certificate, Level 1 national diploma, O-level A–C, intermediate apprenticeship)	1	7
Level 3 (Level 3 NVQ, A-level/AS Level, access to higher education diploma, advanced apprenticeship, international baccalaureate diploma, level 3 award, level 3 certificate, level 3 diploma, level 3 ESOL, level 3 national certificate, level 3 national diploma, tech level)	1	7
Level 4–5 (Level 4–5 NVQs, higher national certificate (HNC), higher national diploma (HND), diploma of higher education (DipHE))	4	27
Level 6 (Degree with or without honors (e.g., BA, BSc)	6	40
Level 7 (Higher degree (e.g., master’s degree (MA, MSc), postgraduate certificate in education (PGCE))	2	13
Not recorded	1	7
Total	15	100 *

* Some percentages do not add up to 100; where numbers have been rounded up to whole numbers or participants have selected more than one answer. ** Based on the levels of qualifications for England, Wales and Northern Ireland [70].

## Data Availability

The data presented in this study are available on request from the corresponding author. The data are not publicly available due to permission from participants not being explicitly sought for this.

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
