# Peer review of "Improving Vitamin D Intake in Young Children—Can an Infographic Help Parents and Carers Understand the Recommendations?"

_nutrients, 2021, doi:10.3390/nu13093140_

Round 1

Reviewer 1 Report

Review: Improving vitamin D intake in young children – can an info-graphic help parents and carers understand the recommendations?

Thank you for the opportunity to review this manuscript titled Improving vitamin D intake in young children – can an info-graphic help parents and carers understand the recommendations? ‘. It is generally well written; however, I do suggest some minor grammatical editing throughout. The results are clearly presented, and the discussion is well articulated with all major strengths clearly identified. I believe this manuscript is suitable for publication after the following recommendations are considered by the authors:  

Introduction

General comment: The introduction is too long. It should focus more on the topic of study.I recommend that it should follow this structure: (4 paragraphs)

-Presentation of vitamin D, synthesis and health consequences.

-Consequences of deficiency on children's health. Situation and status of recommendations in the UK.

-Synthesis lines 129-160.

-Justification and objective lines (161-167) First justification and then objective:

To the researchers’ knowledge, this type of research is the first of its kind, with 164 regards to testing an infographic displaying the current UK infant vitamin D recommen- 165 dations, and thus provides unique information to inform the development of a valuable 166 tool to effectively educate the community around vitamin D in the early years.).  Therefore, this study aims to develop and test the acceptability and understanding of 161 an infographic displaying vitamin D recommendations, with a group of parents or carers 162 of children aged under 5 years, as an intervention to improve vitamin D status in young 163 children.

Material and methos

Line 170: “The infographic was developed in response to findings from previous research”. References.

Line 190-192.Why these stages? “It demonstrates the 190 vitamin D recommendations broken down into the following stages: pregnancy; breast- 191 feeding exclusively (0-6 months); formula feeding (0-6 months); weaning (6 months – 1 192 year) and for the child aged 1 to 4 years. Needs reference.

2.2.1. Participants and recruitment. Decribe your simple of participants.

the sessions were held. Specify

Line 210. Are there exclusion criteria? clarify.

Line 239-240.Who led the sessions How were they recorded? Who transcribed them and how? Please clarify

Line 251. “Reflexivity journaling was also carried out after 6 of 18 each session to help provide context and understanding of the study and to take learning from and improve subsequent sessions”. Who used the journal

Lines 261-270. Who was in charge of coding the categories?

Discussion

Please add a first paragraph with the main results found. And then discuss each result in order.

Highlight the main limitations of your study: qualitative study, most of your sample has a higher level of education, etc.

Conclusion

Summarize the conclusion. The conclusion should be clear and direct.

Reviewer 2 Report

The authors of the manuscript "Improving vitamin d intake in young children - can an infographic help parents and carers understand the recommendtions?"  report in detail the response of the participants to the proposed infographic. Some suggestions:

  1. The methods section is very long and could be summarized
  2. A Page 5, line 229 in the phrase "telephone interviews were conducted by one female researcher" why is the sex of the interviewer specified?
  3. A Page 6, Line 283 better describe what is a "Level 4 qualification"
  4. The results section is very long and could be summarized by putting the part relating to the individual sentences of the participants as supplementary material
  5. Is there a difference between the comments on the infographic based on the age groups of the participants?
  6. Is there a difference between the comments on the infographic based on the school education received by the participants?

Round 2

Reviewer 1 Report

Dear authors, 

Thank you for incorporating all my comments. The paper is well written and interesting. The description of the participants is correct, both in methodology and results.

Thank you.

Reviewer 2 Report

the authors answered the questions